# Benzodiazepines and Mood Stabilizers in Schizophrenia Patients Treated with Oral versus Long-Acting Injectable Antipsychotics—An Observational Study

**DOI:** 10.3390/brainsci13020173

**Published:** 2023-01-20

**Authors:** Ana Aliana Miron, Paula Simina Petric, Andreea Teodorescu, Petru Ifteni, Gabriela Chele, Andreea Silvana Szalontay

**Affiliations:** 1Facultatea de Medicină, Universitatea Transilvania din Brașov, Bulevardul Eroilor nr. 29, 500036 Brașov, Romania; 2Spitalul Clinic de Psihiatrie și Neurologie Brașov, Str. Prundului nr. 7-9, 500123 Brașov, Romania; 3Facultatea de Medicină, Universitatea de Medicină si Farmacie, Grigore T. Popa” Iași, Str. Universității nr. 16, 700115 Iași, Romania; 4Institutul de Psihiatrie _„_Socola” Iași, Şoseaua Bucium nr. 36, 700282 Iași, Romania

**Keywords:** schizophrenia, antipsychotics, mood stabilizers, benzodiazepines

## Abstract

Schizophrenia is a chronic, invalidating, and polymorphic disease, characterized by relapses and remission periods. The main treatment option in schizophrenia are antipsychotics, administered as an oral or as a long-acting injectable (LAI) formulation. Although international guidelines rarely recommend it, mood stabilizers (MS) and/or benzodiazepines (BZD) are frequently prescribed as adjunctive therapy in schizophrenia patients for various reasons. This is an observational, cross-sectional study including stabilized schizophrenia patients. A total of 315 patients were enrolled. Of these, 77 patients (24.44%) were stabilized on LAIs and 238 (75.56%) patients on oral antipsychotics (OAP). Eighty-four patients (26.66%) had concomitant treatment with MS and 119 patients (37.77%) had concomitant benzodiazepine treatment. No statistical significance was observed in MS or BZD use between LAIs and OAPs. In total, 136 patients (43.17%) were stabilized on antipsychotic monotherapy. Our study shows that the long-term use of benzodiazepines and mood stabilizers remains elevated among stabilized schizophrenia patients, regardless of the antipsychotic formulation (oral or LAI). Patients receiving second-generation LAI antipsychotics (SGA-LAI) seem to be more likely to be stabilized on monotherapy compared to those receiving oral antipsychotics. Further randomized controlled trials are necessary in order to clarify the benefits of the current drug polypharmacy trends.

## 1. Introduction

Schizophrenia is a chronic, invalidating, and polymorphic disease characterized by positive symptoms (delusions, hallucinations, disorganized speech and/or behavior), negative symptoms (apathy, social isolation, diminished affect), motivational dysfunction, and cognitive impairment [1]. In most cases, its evolution is characterized by relapses and remission periods. Psychotic relapses are frequent [2] and may have devastating consequences [3]. In addition to the main and defining psychotic symptoms, patients with schizophrenia also frequently present other symptoms, such as anxiety [4], poor impulse control [5], or affective symptoms [6]. The main treatment option in schizophrenia are antipsychotics. Chronic antipsychotic treatment may be administered as an oral formulation or as a long-acting injectable (LAI) formulation. The goals of antipsychotic treatment are symptom remission, relapse prevention, and ultimately maintaining or even improving patients’ global functionality. Multiple studies have shown that LAI antipsychotics have benefits in reducing relapses, hospitalizations, general morbidity, and mortality in schizophrenia patients [7,8,9]. In spite of the evidence, choosing an oral or an LAI antipsychotic as maintenance treatment seems to remain a challenge, most likely because of concerns regarding their safety versus oral antipsychotics (OAP) and doubts regarding the best initiation time or the benefits outside adherence improvement [10,11]. Multiple adjuvant medication used concurrently with an antipsychotic can be another barrier to starting an LAI.

Drug polypharmacy is to be avoided, as recommended by treatment guidelines [12]. Reports show that at least 20% of individuals with schizophrenia do not experience a substantial response to monotherapy with antipsychotics [13]. Schizophrenia patients are frequently prescribed adjunctive treatments such as benzodiazepines (BZD), mood stabilizers (MS), antidepressants, or hypnotics for a better control of the symptoms [14,15]. Benzodiazepines are often prescribed for sleep disturbances, anxiety, or hostile behavior. Some studies support the combination of antipsychotics and benzodiazepines as beneficial for positive and negative symptoms, as well as catatonia or adverse reactions to antipsychotic medications. This clinical phenomenon seems to be associated with the gamma-aminobutyric acid (GABA-ergic) activity that is believed to be disrupted in the schizophrenia and direct benzodiazepines effect on GABA-A receptors [16]. Benzodiazepines are believed to reduce presynaptic dopamine release at the mesolimbic level and delay postsynaptic adaptation of dopaminergic neurons to antipsychotics, thus potentiating the action of antipsychotics in resistant schizophrenia. Benzodiazepines also act on mesocortical regions where antipsychotics are less effective and where there is a particular sensitivity to stress. The association of antipsychotics and benzodiazepines is particularly useful in resistant patients or in patients with residual symptoms such as anxiety or emotional flattening [16,17]. Benzodiazepines’ effectiveness in schizophrenia might also be explained by stress being one mediator of relapse in these patients [18]. In addition, inhibition of dopamine neurotransmission through gamma-aminobutyric acid-enhancing activity may provide a direct antipsychotic effect [19]. On the other hand, some authors found that benzodiazepines are inferior to antipsychotics on longer-term global outcomes and that evidence regarding the addition of benzodiazepines to antipsychotics is conflicting [20]. One cohort study concluded that benzodiazepines use is associated with an increased risk of readmission [21]. Furthermore, some authors suggest that benzodiazepine use significantly increases the mortality risk in schizophrenia patients, both as monotherapy and as adjunctive to antipsychotics [22]. Important adverse effects include sedation, cognitive impairment, behavioral disinhibition, exacerbation of psychotic symptoms, and a potential for abuse, withdrawal, and dependence [18].

Although international guidelines rarely recommend it, mood stabilizers are also frequently prescribed as adjunctive therapy in schizophrenia patients, especially in treatment-resistant schizophrenia [23]. Usually, lithium and valproic acid (VPA) are indicated in the treatment of bipolar disorder. Lately, their use has expanded to other neuropsychiatric disorders [24] and schizophrenia [25,26]. One research found that 14.1% of first-episode schizophrenia patients startedMS use within three years from diagnosis. Female gender, younger age, and benzodiazepine use were associated with higher risk of antidepressants (AD) and MS initiation, while the number of previous psychoses was associated with decreased risk of AD and increased risk of MS initiation [27]. Another study reported that MS were prescribed with antipsychotics in 13.6% of schizophrenia patients and high doses of MS were given in patients with a less favorable illness course, more behavioral disorganization, poorer functioning, and higher antipsychotic doses [28]. A literature review shows that lithium has some effects on affective symptoms and holds the best evidence for an anti-aggressive effect; carbamazepine also has anti-aggressive potential in schizophrenia patients. The same review concluded that valproate has not shown a consistently positive effect and that lamotrigine has demonstrated a beneficial effect in one placebo-controlled study [29]. A 2003 study reported significant improvement in patients treated with an association of olanzapine or risperidone and VPA compared to olanzapine or risperidone monotherapy [30]. A recent Finish based cohort study concluded that mood stabilizers use was associated with a 12% decreased risk of admission due to psychosis. A lower risk of psychosis hospitalization was associated with the use of lithium, valproic acid, and lamotrigine compared with non-users, but not with carbamazepine [31]. Tseng et al. reported significant benefits of valproate augmentation therapy in schizophrenia [32].

### Aims

Our research’s primary objective was to investigate the concomitant use of benzodiazepines and mood stabilizers among stabilized schizophrenia patients. Our secondary objective was to verify whether there are any significant differences in BZD and MS use in oral antipsychotics (OAPs)- versus LAI-treated patients.

## 2. Materials and Methods

### 2.1. Data Source

This is an observational study, extended over 12 months, that included patients diagnosed with schizophrenia according to the DSM-5 criteria. Research was conducted in “Spitalul Clinic de Psihiatrie și Neurologie Brașov”, a medical academic center from Brasov, Romania, with 100 beds for acute patients and 315 beds for long-term admission patients. All subjects signed an informed consent for participating in the study. Data was collected by board-certified psychiatrists from the paper files of the patients from the hospital and psychiatric ambulatory settings. The database obtained included age, gender, and drug product information (antipsychotic type, formulation and dosage, mood stabilizer type and dosage, benzodiazepine type and dosage). The study was approved by the Hospital Ethics Committee (approval no. 1/02.03.2021).

### 2.2. Study Design

Using a cross-sectional design, the study had an enrollment period between 1 June 2021 and 31 May 2022. Inclusion criteria were: outpatients, medically ensured, aged over 18 years, diagnosed with schizophrenia according to the DSM-5 criteria, stabilized on antipsychotic treatment, and without admission for psychotic episodes in the past 6 months. No upper age limit was set for the study population. All patients’ previous admissions were verified in the hospitals’ database. All patients were treated with oral or long-acting injectable antipsychotics. Exclusion criteria were: diagnosis of schizoaffective disorder and patients without antipsychotic treatment.

### 2.3. Statistical Analysis

Results were analyzed using the SPSS program version 20.00. The adjusted odds ratio (AOR) with 95% CI was calculated and *p*-values lower than 0.05 were considered as statistically significant. To compare means, we used a *t*-test to calculate statistical significance. To compare proportions, the Chi-square test was used. The multivariable logistic regression was considered to indicate a significant association.

For the calculation of equivalent doses of chlorpromazine, we used equivalence tables [33,34], the consensus method of Gardner et al. [35], and the classical mean dose method [36]. For atypical LAIs, conversion was firstly calculated to the equivalent oral dose, then to chlorpromazine.

## 3. Results

### 3.1. General Findings

From the initial cohort, which comprised 566 schizophrenia patients, 315 patients met the inclusion-exclusion criteria and were enrolled in the study. 77 patients (24.44%) were stabilized on LAI treatment (LAI subgroup) and 238 (75.56%) patients on oral treatment (OAP subgroup); 18 patients (5.71%) were stabilized on combined antipsychotic, LAI, and oral treatment. In the LAI group, 39 patients (50.64%) were stabilized on second-generation LAI antipsychotics (SGA-LAI group) and 38 patients (49.36%) on first-generation LAI antipsychotics (FGA-LAI group). Patients stabilized on SGA-LAI treatment represent 12.38% of all patients in the study. Demographics and treatment characteristics of LAI and OAP populations are summarized in Table 1 [Appendix A.

The mean age in the entire group of patients was 51.71 (±11.39 SD). In the LAI group, the mean age was 52.92 (±12.24 SD), while in the OAP group, the mean age was 51.32 (±11.10 SD). The difference is not statistically significant (*p* = 0.28). Among patients treated with LAI, the SGA-LAI sub-group had a mean age of 48.79 (±11.75 SD), while the FGA-LAI subgroup had a mean age of 57.15 (±11.39 SD). In the SGA-LAI group, the mean age was significantly lower than in the FGA-LAI group (*p* = 0.0022). When comparing the mean ages between the SGA-LAI and OAP groups, the difference was not statistically significant (*p* = 0.19).

Out of the total 315 patients, 130 (41.27%) patients were male. 96 patients (40.34%) in the OAP group and 34 (44.15%) in the LAI group were male. No statistically significant difference was observed between the percentages of male patients receiving LAI or OAP (*p* = 0.55). In the SGA-LAI group, 23 patients (58.97%) were female and 16 (41.03%) male (*p* = 0.11), and in the FGA-LAI group, 20 patients (52.63%) were female and 18 (47.37%) male (*p* = 0.64). No statistically significant difference was observed between the percentages of male patients receiving SGA-LAI or FGA-LAI (*p* = 0.57). Age group distribution in LAI and OAP groups is illustrated in Figure 1 [Appendix A.

The most frequently used SGA-LAI was risperidone (20.77%) and the most used FGA-LAI was flupenthixol (46.75%). The most used OAPs were olanzapine (29.41%) and clozapine (27.73%). Mean doses of antipsychotics are detailed in Table 2 [Appendix A.

### 3.2. Mood Stabilizers

Of the 315 patients, a total of 84 patients (26.66%) were treated with MS. All 84 patients received sodium valproate. There was no concomitant treatment with carbamazepine and lamotrigine. Distribution of cases according to concomitant treatments is illustrated in Figure 2 [Appendix A.

Sixteen patients (20.77%) in the LAI group and 68 patients (28.57%) in the OAP group received MS concomitant treatment. MS prevalence in the LAI group was 0.20 and in the OAP group 0.28 (OR = 0.65, 95% CI, −3.55% to 17.54%, *p* = 0.16). No significant difference was observed in the LAI versus OAP groups.

In the SGA-LAI population, nine patients (23.07%) received MS treatment, and in the FGA-LAI population, seven patients (18.42%) were identified. Percentage comparison calculation found that the differences are not statistically significant either between the SGA-LAI versus the FGA-LAI population (p = 0.61; OR = 1.32) or between the SGA-LAI versus the OAP population (*p* = 0.47; OR = 0.75).

### 3.3. Benzodiazepines

A total of 119 patients (37.77%) had benzodiazepine treatment. Eight patients (2.53%) had concomitant treatment with more than one benzodiazepine. BZD prevalence in the LAI group was 0.376 and in the OAP group 0.378 (OR = 0.99, 95% CI, −12.43% to 12.01%, *p* = 0.97). No significant difference was observed in the LAI versus the OAP group.

BZD prevalence was 0.28 in the SGA-LAI group and 0.47 in the FGA-LAI group. Although we noted that there were fewer patients with concomitant benzodiazepine treatment in the SGA-LAI group compared to the FGA-LAI group, the difference is not statistically significant (OR = 0.43, 95% CI, −2.38% to 38.45%, *p* = 0.08). Comparison between SGA-LAI and OAP groups did not reveal a significant difference in BZD use (OR = 0.64, 95% CI, −7.05% to 22.85%, *p* = 0.24). BZD types used in the study groups are detailed in Table 3 [Appendix A.

The most frequently used BZD was diazepam (15.58% in the LAI group, 18.06% in the OAP group), followed by lorazepam (14.28% in the LAI group, 20.58% in the OAP group). No significant difference was noted in the use of specific benzodiazepines in LAI versus OAP groups (*p* = 0.61 for diazepam; *p* = 0.22 for lorazepam). In the SGA-LAI group, we checked for differences in the LAI formulations versus their oral correspondent. Patients’ concomitant treatment according to the type and formulation of antipsychotic used is detailed in Table 4 [Appendix A.

No statistical significance was observed in MS or BZD use between LAI formulations and their oral correspondents. Among SGA-LAI antipsychotics, risperidone has the highest percentages of associated MS (25%) and associated BZDs (37.5%). In the OAP group, risperidone also had the highest percentage of associated MS (35.48%) and BZDs (38.70%). In the case of concomitant combined treatment (MS and BZD), the highest percentage of patients was registered for aripiprazole LAI (22.2%) and oral risperidone (19.35%).

### 3.4. Monotherapy versus Polytherapy

Thirty-nine patients were stabilized on a combination of two or more antipsychotics (12.38%). In the SGA-LAI group, we identified nine patients (23.07%); in the FGA-LAI group, nine patients (23.68%); and in the OAP group, we identified 21 patients (8.82%) treated with combinations of antipsychotics. group No significant difference was found between SGA-LAI and FGA-LAI group (*p* = 0.94). However, we found statistical significance when comparing the percentages in the total LAI group (18 patients, 23.37%) with the OAP group (*p* = 0.0008), as well as when comparing SGA-LAI with OAP (*p* = 0.008) and FGA-LAI with OAP (*p* = 0.006).

The highest percentage of monotherapy in the OAP group was observed for olanzapine (45.71%), followed by clozapine (40.90%). There was no statistically significant difference in the monotherapy cases in the LAI versus OAP groups (44.15% versus 42.85%, *p* = 0.84).

Monotherapy cases in the SGA-LAI group versus their oral correspondents are detailed in Table 5 [Appendix A.

No statistical significance was revealed when comparing the SGA-LAI subgroup with their oral correspondents. The highest percentage of monotherapy cases is observed with olanzapine LAI (80%), followed by aripiprazole LAI (55.55%).

## 4. Discussion

To our knowledge, this is the first study comparing the use of benzodiazepines and mood stabilizers in oral- versus LAI-stabilized schizophrenia patients. Our results show that 12.38% of patients were stabilized on SGA-LAIs, percentage that is similar to other reportsIn USA, 4–28% of patients requiring antipsychotic treatment receive a LAI [37,38,39]. A recent Swiss study showed that 49% of patients were eligible for SGA-LAI administration, but only 28.1% received this type of treatment, and only 15.5% of schizophrenia patients were prescribed SGA-LAIs [40].

In the SGA-LAI group in our research, the mean age is significantly lower than in the FGA-LAI group (*p* = 0.0022), but not than in the OAP group (*p* = 0.19). Our result is consistent with other studies, which have shown that SGA-LAIs are more likely to be prescribed to younger patients, but that after the age of 40, patients with schizophrenia receive significantly more FGA-LAIs than SGA- LAIs [41]. On the other hand, a US study showed that, on average, patients initiated on SGA-LAI were younger than those initiated on OAP (42.2 vs. 44.8 years; *p* < 0.001), while mean age seemed similar between patients with FGA-LAI and OAP [42]. Our study group consists of patients already stabilized on antipsychotic treatment, not at initiation; this could explain the fact that mean age in all our subgroups highlighted is slightly higher than reported in the literature, as well as the significantly higher age of patients in the FGA-LAI group.

No statistically significant difference was noted between percentages of male or female patients stabilized on SGA-LAI, FGA-LAI, or OAP treatment. Also, our results did not show any predilection for a certain antipsychotic type or formulation for men or women. Some studies suggest that a higher proportion of patients initiated on OAP are women compared to FGA-LAI patients or SGA-LAI patients [42].

Although some studies indicate paliperidone as the most prescribed SGA-LAI [40,43], our research points to risperidone (41.04%) as the most used SGA-LAI. Flupenthixol (94.73%) was the most prescribed FGA-LAI. In the OAP group, olanzapine is most frequently prescribed (29.41%), followed by clozapine (27.73%) and risperidone (13.02%). A recent literature review reports, however, that the most commonly used OAPs in schizophrenia are olanzapine (up to 50.9%), risperidone (up to 40.0%), and quetiapine (up to 30.7%) [44]. Clozapine, the gold standard for treatment-resistant schizophrenia, was used in nearly a third of OAP patients in our study; indeed, the literature reports that up to 30% of patients have treatment-resistant schizophrenia [45].

Our data showed that only for olanzapine the use is significantly lower in the LAI formulation compared to the oral formulation. Interpretation of this result requires caution due to the small number of patients stabilized on olanzapine LAI. We believe, however, that this reflects the reality of olanzapine LAI prescriptions and it might be explained by several reasons. Olanzapine LAI is not frequently prescribed because of fears regarding post-injection delirium-sedation syndrome and because protocols impose at least 3 h of observation after administration. In addition, during the COVID-19 pandemic, a considerable decrease in the initiation of LAI treatments was reported [46] and due to nation-wide pandemic-related restrictions, most patients on olanzapine LAI were switched to oral formulation [47].

Nearly a third of all patients (26.66%) had concomitant treatment with a MS. In the OAP group, the percentage of patients (28.57%) was higher compared to the LAI group (20.77%), but without statistical significance. Surprisingly, in the SGA-LAI group, MSs were used in a higher proportion compared to the FGA-LAI group (23.07% versus 18.42%), but no statistical significance was identified. The literature reports a smaller percentage of MS-associated use, varying between 13.6% and 14.1% [27,28,47]. Thus, we note a rather high co-prescription of valproate and antipsychotics, which is lacking solid scientific substantiation and despite the fact that valproate is a known teratogenic agent [48]. Recent research has shown that over a third of patients co-treated with valproate have off-label prescriptions [49]. Although some authors support benefits of combined antipsychotic-MS treatment in preventing hospitalizations for psychotic episodes [31,32], other data suggest that valproate does not have a consistent positive effect on affective or aggressive symptoms in schizophrenia [29]. A Cochrane review evaluating the effectiveness of valproic acid as an add-on to antipsychotics found limited evidence of better clinical response, which was lost when low-quality studies were excluded. However, there was some evidence that valproic acid may be effective in controlling excitement and aggression [50]. As for other MSs, the literature does not appear supportive of long-term use. Data regarding carbamazepine suggests that it cannot be routinely recommended for the treatment of schizophrenia [51]. Some authors suggest limited efficacy for lamotrigine [52] and inconclusive evidence regarding the effectiveness of lithium combined with an antipsychotic [25].

Over a third of the total number of patients (37.77%) received concomitant benzodiazepine treatment. Although not statistically significant, we did notice that in the SGA-LAI group, the percentage was lower than the FGA-LAI group (28.2% versus 47.36%). Most often prescribed among all groups were diazepam and lorazepam. A similar cross-sectional study investigated the use of benzodiazepines among long-term hospitalized schizophrenia patients. The research found that 6.6% of the schizophrenia patients received regular long-term BZD treatment and 2.5% were prescribed BZDs as needed. Most prescribed BZDs were lorazepam, diazepam, clonazepam, and temazepam. The same research reported only four patients on LAI formulations, but the authors did not specify which type and dose and did not further analyze differences in the oral versus LAI subgroups [53]. Although authors support the role of adjunctive benzodiazepine treatment as beneficial in schizophrenia patients [16,17], a meta-analysis clearly points out that there is no evidence of improvement in psychotic symptoms by adding benzodiazepine medication in schizophrenia [54]. Another review showed improvement only in the first 30 min after the association of BZD to anantipsychotic, while sedation as a desired effect was significant in the first 30 and 60 min evaluations. This might lead to the conclusion that BZDs are only suitable in the acute management of psychotic episodes [55]. Of note, BZD use might generate significant side effects, such as cognitive impairment [56], abuse and dependence [57], and increased mortality risk [22].

A question arises: “Why add mood stabilizers and/or benzodiazepines in the maintenance treatment for schizophrenia?” It is a known fact that multi-drug regimens generate lower adherence rates [58]. A possible answer could be that antipsychotics are ineffective in controlling all symptoms of the disease. Some authors suggest that antipsychotic monotherapy is indeed insufficient for up to 20% of patients with schizophrenia [13]. A recent study shows that valproate is used in patients with schizophrenia to treat affective symptoms (62%), disease-refractory symptoms (10%), aggression (6%), seizures (as prevention in those treated with clozapine—6%, or as treatment—6%), or with unclear reasons (16%) [49].

Another reason could be that antipsychotics are improperly dosed. Our results did not show dosages lower than 95% of the effective dose in the study group [59] and furthermore, with the exception of paliperidone, no statistically significant difference was found between mean doses of SGA-LAI compared to their oral correspondents, so in our case, this hypothesis is not valid. The equivalent dose of chlorpromazine in our study was the highest with quetiapine, which is most likely explained by the dose-effect relationship and the 600–800 mg recommendations shown to be effective with quetiapine [60].

The need for concomitant treatment could be explained by treatment-resistant schizophrenia. The literature shows that up to 30% of patients may have treatment-resistant schizophrenia [45]. In our study, a total of 66 patients were treated with clozapine (20.95%), 53 of which (16.82%) had only clozapine as antipsychotic treatment, while 13 (4.12%) had combined treatment (clozapine and another antipsychotic). Three patients had a clozapine dose under 100 mg/day, in which case we can assume that it was used for reasons other than as an antipsychotic, for instance as a hypnotic [61]. Assuming that all patients treated with clozapine have treatment-resistant schizophrenia, their percentage is still lower than reported by the literature. Therefore, we consider that at least a part of the patients that were co-treated with mood stabilizers and/or benzodiazepines might in fact have treatment-resistant schizophrenia.

Concomitant treatments could also be prescribed to treat some side effects of antipsychotics. Benzodiazepines in particular are commonly used for the treatment of extrapyramidal adverse effects, which can occur in up to 37% of patients [62]. Evidence shows that patients under FGA-LAI treatment, typical oral drugs, risperidone, and amisulpride have a significantly higher risk of developing extra- pyramidal symptoms (EPS) [63]. This could explain our findings showing that the FGA-LAI group had a higher percentage of BZD use compared to SGA-LAI, and that risperidone patients, regardless of the formulation, had the highest rate of concomitant benzodiazepine treatment. Our results also indicated that in the SGA-LAI group, the percentage of benzodiazepine users was lower than in the OAP group, which might suggest a lower percentage of extrapyramidal side effects. This is opposite to other studies showing that extrapyramidal side effects occur significantly more often in the SGA-LAI group than in the oral SGA group (RR, 1.61; 95% CI) [64]. Given the relatively small group of SGA-LAI patients in our study, caution is warranted in interpreting this result. In addition, benzodiazepines are quite commonly used as a symptomatic treatment of anxiety or sleep disturbances, which occur in a relatively high percentage of patients with schizophrenia [65]. As for valproic acid, it can be used as an anticonvulsant treatment or as seizure prophylaxis in clozapine-treated patients [49]. Our data indicates that 12 patients (18.18%) stabilized on clozapine also had concomitant treatment with valproate. The reported incidence of seizures associated with clozapine is between 1% and 4.4% [66], so this hypothesis can only explain a small part of the concomitant valproate treatment.

We noticed that 12.38% of all patients were stabilized on a combination of two or more antipsychotics. Surprisingly, patients in the LAI group have a significantly higher rate of antipsychotic polytherapy compared to the OAP group (*p* = 0.0008). Our results seem to align with the growing trend of drug polypharmacy in schizophrenia; some authors report that up to 30% of patients are treated with two or more antipsychotics [67]. Potential benefits have been reported, especially for particular combinations such as aripiprazole and clozapine, in reducing the hospitalization risk [68]. Less than half of our total group of patients was stabilized on antipsychotic monotherapy (43.17%). The highest percentage in the LAI group is observed for olanzapine LAI (80%), followed by aripiprazole LAI (55.55%), but due to the small size of the samples, the results cannot be extrapolated. In the OAP group, the highest percentage of monotherapy was identified for olanzapine (45.71%), followed by clozapine (40.90%). Zhang et al concluded that olanzapine monotherapy is superior to olanzapine polypharmacy, and to any type of risperidone treatment (monotherapy or polypharmacy), in terms of discontinuation rates and time to discontinuation [69]. Based on accumulating evidence, some authors even recommend that the therapeutic guidelines should revise their current discouraging policy regarding polypharmacy in the maintenance treatment of schizophrenia.

As any observational study, our research has some limitations. The number of patients in the LAI group is relatively small but this actually reflects their real-life underutilization. The cross-sectional design of the study cannot capture the subsequent modification of treatment, which prevents the evaluation of treatment changes over time. Another limitation might be that this study focused on the use of benzodiazepines and mood stabilizers, so the concomitant use of other medications (z-drugs, antidepressants) was not thoroughly investigated. In Romania, the availability of antipsychotics and the reimbursement policy of these drugs for schizophrenia make our results easily generalizable at the country level, which represents a strength of our research. On the other hand, particularities of other countries’ medical systems and restrictions of national guidelines make our results difficult to superimpose worldwide.

## 5. Conclusions

Our study shows that long-term use of benzodiazepines and mood stabilizers among schizophrenia patients remains elevated. This phenomenon is also observed for LAI formulation. We emphasize the fact that benzodiazepines and mood stabilizers should be prescribed, when needed, for only short periods of time, due to their side effects and since their long-term use is not recommended by clinical guidelines. Polypharmacy can also be an obstacle to LAI initiation. Further randomized controlled trials are necessary in order to clarify the efficacy and safety of benzodiazepines and mood stabilizers use for long periods of time in schizophrenia.

## Figures and Tables

**Figure 1 brainsci-13-00173-f001:**
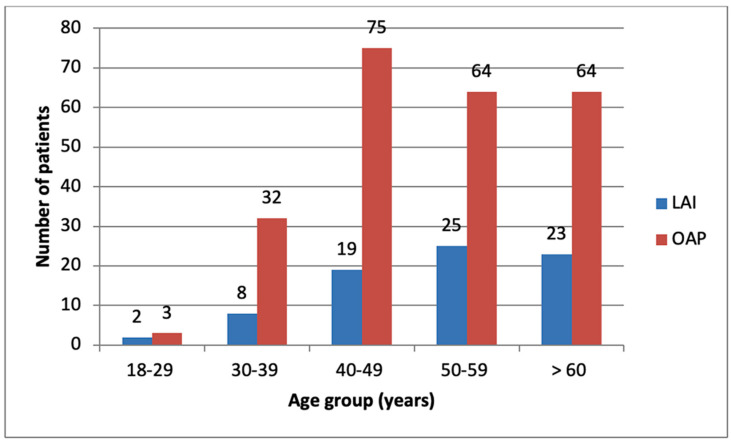
Age group distribution of LAI and OAP groups.

**Figure 2 brainsci-13-00173-f002:**
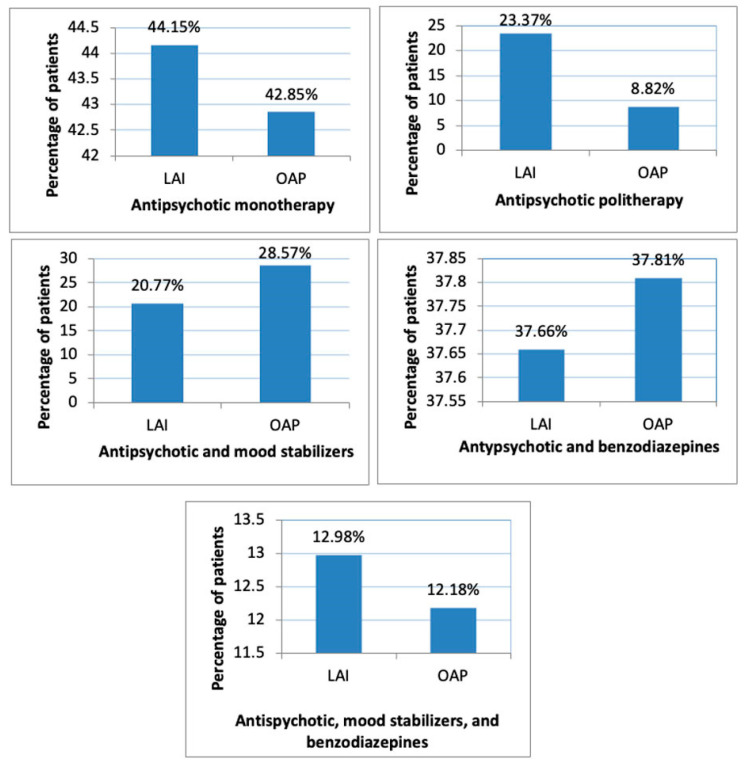
Concomitant treatment in LAI versus OAP groups.

**Table 1 brainsci-13-00173-t001:** Demographics and treatment characteristics of the study population.

Parameters		LAIs	OAPs	*p*-Value
Number of patients (N, %)		77 (24.44%)	238 (75.56%)	*p* < 0.0001
Male gender (N, %)		34 (44.15%)	96 (40.34%)	*p* = 0.55
Mean age (±SD)		52.92 (±12.24 SD)	51.32 (±11.10 SD)	*p* = 0.28
Patients receiving BZD (N, %)	Total	29 (37.66%)	90 (37.81%)	*p* = 0.98
	Male	11 (37.93%)	34 (37.77%)	*p* = 0.98
	Female	18 (62.07%)	56 (62.23%)	*p* = 0.93
Patients receiving MS (N, %)	Total	16 (20.77%)	68 (28.57%)	*p* = 0.17
	Male	8 (50%)	32 (47.05%)	*p* = 0.83
	Female	8 (50%)	36 (52.95%)	*p* = 0.83
Patients receiving both MS and BZD (N, %)	Total	10 (12.98%)	29 (12.18%)	*p* = 0.85
	Male	5 (50%)	15 (51.72%)	*p* = 0.92
	Female	5 (50%)	14 (48.28%)	*p* = 0.92
Antipsychotic type	OLZ (N, %)	5 (6.49%)	70 (29.41%)	*p* ˂ 0.0001
	RIS (N, %)	16 (20.77%)	31 (13.02%)	*p* = 0.09
	PAL (N, %)	9 (11.68%)	17 (7.14%)	*p* = 0.20
	ARI (N, %)	9 (11.68%)	24 (10.08%)	*p* = 0.69
	QUE (N, %)	-	24 (10.08%)	-
	AMI (N, %)	-	20 (8.40%)	-
	ZIP (N, %)	-	1 (0.42%)	-
	HAL (N, %)	0	22 (9.24%)	-
	FLX (N, %)	36 (46.75%)	-	-
	ZUC (N, %)	2 (2.59%)	-	-
	LEV (N, %)	-	3 (1.26%)	-
	TIA (N, %)	-	1 (0.42%)	-
	CLZ (N, %)	-	66 (27.73%)	-

Legend: OLZ = olanzapine; RIS = risperidone; PAL = paliperidone; ARI = aripiprazole; QUE = quetiapine; AMI = amisulpride; ZIP = ziprasidone; HAL = haloperidol; FLX = flupenthixol; ZUC = zuclopenthixol; LEV = levomepromazine; TIA = tiapridal; CLZ = clozapine.

**Table 2 brainsci-13-00173-t002:** Mean antipsychotic dose and chlorpromazine equivalents.

Antipsychotic (Type, Formulation)			Number of Cases	Mean Dose (mg)	Oral Dose Equivalent (mg)	Chlorpro-Mazine Equivalent (mg)	*p*-Value
olanzapine	LAI		5	480 (±164.31)	16 (±5.47)	320	*p* = 0.66
	OAP		70	15 (±5)	15 (±5)	300	
risperidone	LAI		16	76.56 (±24.94)	3.06 (±0.99)	306	*p* = 0.21
	OAP		31	3.58 (±1.50)	3.58 (±1.50)	358	
aripiprazole	LAI		9	400	20	266.66	*p* = 0.22
	OAP		24	16.875 (±7.49)	16.875 (±7.49)	225	
paliperidone	LAI		9	180.55 (±152.97)	9.66 (±1.32)	483	*p* = 0.005
	OAP		15	7.4 (±1.91)	7.4 (±1.91)	370	
quetiapine	LAI		-	-	-	-	NA
	OAP		24	491.66 (±224.89)	-	655	
amisulpride	LAI		-	-	-	-	NA
	OAP		20	500 (±247.08)	-	290	
ziprasidone	LAI		-	-	-	-	NA
	OAP		1	120	-	200	
haloperidol	LAI		-	-	-	-	NA
	OAP		22	3.52 (±1.61)	-	176	
flupenthixol	LAI		36	35 (±8.78)	-	100	NA
	OAP		-	-	-	-	
zuclopenthixol	LAI		2	200	-	100	NA
	OAP		-	-	-	-	
levomepromazine	LAI		-	-	-	-	NA
	OAP		3	54.16 (±7.21)	-	54.16	
tiapridal	LAI		-	-	-	-	NA
	OAP		1	100	-	100	
clozapine	LAI		-	-	-	-	NA
	OAP	Total	66	272.72 (±126.51)	-	272.72	
		clozapine monotherapy	53	297.64 (±128.56)		297.64	

**Table 3 brainsci-13-00173-t003:** Benzodiazepine types.

	Diazepam	Alprazolam	Lorazepam	Clonazepam	Bromazepam	Nitrazepam	Cinolazepam
SGA-LAI (N, %)	4 (10.25%)	1 (2.56%)	3 (7.69)	2 (5.12%)	1 (2.56%)	1 (2.56%)	1 (2.56%)
FGA-LAI (N, %)	8 (21.05%)	0	8 (21.05%)	1 (2.63%)	1 (2.63%)	0	1 (2.63%)
OAP (N, %)	43 (18.06%)	4 (1.68%)	49 (20.58%)	16 (6.72%)	2 (0.84%)	0	1 (0.42%)

**Table 4 brainsci-13-00173-t004:** Concomitant treatment by antipsychotic type.

Antipsychotic Type	Formulation	MS (N, %)	*p*-Value	BZD (N, %)	*p*-Value	MSs and BZDs (N, %)	*p*-Value
olanzapine	LAI (N = 5)	1 (20%)	*p* = 0.77	0	*p* = 0.10	0	*p* = 0.42
	OAP (N = 70)	18 (25.71%)		25 (35.71%)		8 (11.42%)	
risperidone	LAI (N = 16)	4 (25%)	*p* = 0.47	6 (37.50%)	*p* = 0.93	3 (18.75%)	*p* = 0.96
	OAP (N = 31)	11 (35.48%)		12 (38.70)		6 (19.35%)	
aripiprazole	LAI (N = 9)	2 (22.22%)	*p* = 0.69	2 (22.22%)	*p* = 0.54	2 (22.22%)	*p* = 0.49
	OAP (N = 24)	7 (29.16%)		8 (33.33%)		3 (12.50%)	
paliperidone	LAI (N = 9)	2 (22.22%)	*p* = 0.57	3 (33.33%)	*p* = 0.73	1 (11.11%)	*p* = 0.70
	OAP (N=15)	5 (33.33%)		4 (26.66%)		1 (6.66%)	

**Table 5 brainsci-13-00173-t005:** Monotherapy cases in the SGA-LAI group versus oral correspondents.

Antipsychotic (Type, Formulation)		Number of Patients	Patients Stabilized on Monotherapy (N, %)	*p*-Value
olanzapine	LAI	5	4 (80%)	*p* = 0.18
	OAP	70	32 (45.71%)	
risperidone	LAI	16	8 (50%)	*p* = 0.24
	OAP	31	10 (32.25%)	
aripiprazole	LAI	9	5 (55.55%)	*p* = 0.25
	OAP	24	8 (33.33%)	
paliperidone	LAI	9	4 (44.44%)	*p* = 0.83
	OAP	15	6 (40%)	

## Data Availability

Data was retrieved from the paper and electronic documents of the patients. The datasets used and/or analyzed during the current study are available from the corresponding author upon reasonable request.

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
