# Peer review of "Benzodiazepines and Mood Stabilizers in Schizophrenia Patients Treated with Oral versus Long-Acting Injectable Antipsychotics—An Observational Study"

_brainsci, 2023, doi:10.3390/brainsci13020173_

Round 1

Reviewer 1 Report

I would like to appreciate your efforts for presenting the research results. 

However, the methodology section needs more clarification such as sample size calculation, the sample selection process, the sample setting includes the out-pateints setting what about the in-pateints setting?

Also, the the  the ethical approval not clear? can the author provide more details about the institute provide the approval and the ethical number.

Author Response

Q1. The methodology section needs more clarification such as sample size calculation, the sample selection process, the sample setting includes the out-patients setting what about the in-patients setting?

A1. Thank you for your question. Sample size and sample selection process- inclusion and exclusion criteria- are detailed in the Data source and Study Design sections of the manuscript (Rows112-129). As per your suggestions, some clarifications were added; patient’s previous admissions were verified in the hospital database (Row124, Rows 126-127). This study was focused on stabilized schizophrenia outpatients, aiming to investigate the real-life prescribing trends. Concomitant medication in stabilized schizophrenia inpatients represents the focus of a different research.

Q2. Also, the ethical approval not clear? can the author provide more details about the institute provide the approval and the ethical number.

A2. Thank you for your remark. Ethical approval details and approval number were added (Rows120-121).

Reviewer 2 Report

This article gives a detailed insight into the antipsychotic prescription in patients with schizophrenia. Especially, it was considered the antipsychotic formulations, their combinations, and add-on therapy of mood stabilizers or benzodiazepines.

However, there are some obstacles that I would like to point out and suggest a major revision. 

Specific comments:

-       The Title should reflect the major findings, in this way could imply that it is a review. 

-       The upper age limit was not specified as an inclusion criterion (<65?).

-       The Results section could be divided into subsections, with the subtitles reflecting the major findings. Clearly state if other psychotropic drugs were also prescribed (antidepressants, z-hypnotics, or stabilizers such as pregabalin, etc.).

-       The Discussion section could be started with the most prominent findings. Although it was considered in the Introduction, these Results must be also discussed regarding the use of stabilizers and benzodiazepines, especially those cases of polypharmacy. The authors could discuss more thoroughly the receptor profile and mechanism of action of each antipsychotic that is related to the potential of sedation and other clinically wanted or unwanted effects. 

-       As a limitation of this study, the availability of psychotropic drugs in every country and the restriction of national guidelines must be mentioned. 

-       Try making those figures with presented percentages more elegant. 

Author Response

Q1. The Title should reflect the major findings, in this way could imply that it is a review. 

A1. Thank you for this suggestion. The title was modified accordingly (Rows 2-5).

Q2. The upper age limit was not specified as an inclusion criterion (<65?).

A2. Very good observation. For this study, in order to have a better reflection of the real- life patient, we chose not to set an upper limit as inclusion criterion. This specification was added in the manuscript (Rows 126-127).

Q3.  The Results section could be divided into subsections, with the subtitles reflecting the major findings.

A3. Thank you for your suggestion. The Results section was divided accordingly, in 4 sub-sections (General findings- Row 140, Mood stabilizers- Row179, Benzodiazepines- Row 197, Monotherapy versus Polytherapy- Row 223)

Q4. Clearly state if other psychotropic drugs were also prescribed (antidepressants, z-hypnotics, or stabilizers such as pregabalin, etc.).

A4. Very good observation. This study was focused on benzodiazepines and mood stabilizers long- term use in stabilized schizophrenia patients. Concomitant use of other psychotropics was not thoroughly investigated, and as per your suggestion, this was added as a limitation to the study (Rows 386-388).

Q5. The Discussion section could be started with the most prominent findings. Although it was considered in the Introduction, these Results must be also discussed regarding the use of stabilizers and benzodiazepines, especially those cases of polypharmacy. The authors could discuss more thoroughly the receptor profile and mechanism of action of each antipsychotic that is related to the potential of sedation and other clinically wanted or unwanted effects. 

A5. Thank you for your suggestion. Results and Discussion section were updated. We focused on emphasizing the high percentages of benzodiazepine and mood stabilizers prescriptions, in both oral and LAI stabilized patients, as well as polypharmacy, which we consider alarming since current guidelines do not recommend it and since generally multi- drugs regimens generate lower rates of adherence (Marcum ZA, Gellad WF. Medication adherence to multidrug regimens. Clin Geriatr Med. 2012 May;28(2):287-300. doi: 10.1016/j.cger.2012.01.008. PMID: 22500544; PMCID: PMC3335752.)

Q6.  As a limitation of this study, the availability of psychotropic drugs in every country and the restriction of national guidelines must be mentioned

A6. Very good observation. Limitations were completed as per your suggestions (Rows 383- 392)

Q7.  Try making those figures with presented percentages more elegant.

A7. Thank you for this suggestion. Some figures were eliminated from the manuscript, and the remaining figures (Fig 1 and Fig 2) were modified accordingly.

Reviewer 3 Report

Although I found this manuscript quite interesting and current I have to mention that I have severe reserves due to the small sample size of LAI patients (oral AP patients outnumber LAI's 3:1). I am aware that the sample reflects the clinical use of AP's (and other investigated medications) over the period of one year in schizophrenia patients. However, further "diluting" the LAI sample to individual LAI's in my opinion brings out nothing but a view into the clinical practice, which at times can be "messy" (the fact with polytherapy in both groups, be it with AP's, or with MS/BZD). Of course, that is the case with observational studies, which do not require much but also might not offer much in the end. That can be said for the results of this one, as it confirmed some previous findings, but that is just it.

Of minor issues, not DSM-V but DSM-5. Several of the results are presented in the text, in tables and in figures, which is quite redundant.

Author Response

Q1. I have severe reserves due to the small sample size of LAI patients (oral AP patients outnumber LAI's 3:1).

A1. Very good observation. Indeed, the sample size of LAI patients is relatively small, but this is only a reflection of reality, as you have well noticed. This was also mentioned as a limitation to our study (Rows 383- 384).

Q2. Further "diluting" the LAI sample to individual LAI's in my opinion brings out nothing but a view into the clinical practice

A2. Thank you for this remark. The Results section was updated accordingly, and some data was removed from the manuscript. The investigation of particular LAIs, in spite of the small sample size, had the purpose of comparing the prescription habits in LAI versus oral stabilized patients.

Q3. That is the case with observational studies, which do not require much but also might not offer much in the end. That can be said for the results of this one, as it confirmed some previous findings, but that is just it.

A3. Very good observation. A cross- sectional study represents a mirror of what is really happening in the long- term with the patients, of the manner and habits of medication prescribing, to which a large number of factors bring their contribution. At the same time, this is a good way of emphasizing the fact that concomitant benzodiazepines and mood stabilizers should be judiciously used for short periods of time (Rows 395- 399).

Q4. Of minor issues, not DSM-V but DSM-5.

A4. Thank you for your observation. Text was modified accordingly (Rows 114 and 125)

Q5. Several of the results are presented in the text, in tables and in figures, which is quite redundant.

A5. Thank you for your suggestion. Results section of the manuscript was updated accordingly (Figures 3, 4 and 5 were removed).

Round 2

Reviewer 2 Report

The authors need to prepare the manuscript per Journal requirements: the text was moved in this revised version and tables and figures need fine polishing (lines are cutting the numbers, etc.). It was tough to follow the changes because they were not marked.

I want to propose this manuscript for publishing after resolving these technical issues.